# Point-GCC : Universal Self-supervised 3D Scene Pre-training via Geometry-Color Contrast

## ABSTRACT

Geometry and color information provided by the point clouds are both crucial for 3D scene understanding. Two pieces of information characterize the different aspects of point clouds, but existing methods lack an elaborate design for the discrimination and relevance. Hence we explore a 3D self-supervised paradigm that can better utilize the relations of point cloud information. Specifically, we propose a universal 3D scene pre-training framework via **G**eometry-**C**olor **C**ontrast (Point-GCC), which aligns geometry and color information using a Siamese network. To take care of actual application tasks, we design (i) hierarchical supervision with point-level *contrast and reconstruct* and object-level *contrast* based on the novel deep clustering module to close the gap between pre-training and downstream tasks; (ii) architecture-agnostic backbone to adapt for various downstream models. Benefiting from the object-level representation associated with downstream tasks, Point-GCC can directly evaluate model performance and the result demonstrates the effectiveness of our methods. Transfer learning results on a wide range of tasks also show consistent improvements across all datasets. *e.g.*, new state-of-the-art object detection results on SUN RGB-D and S3DIS datasets. Codes will be released on Github.

## CCS CONCEPTS

• **Computing methodologies** → *Unsupervised learning*; **Scene understanding**; 3D imaging.

## KEYWORDS

3D scene point cloud; 3D self-supervised learning; deep clustering

## 1 INTRODUCTION

3D Self-supervised learning (SSL) has received abundant attention recently because of remarkable improvement on various downstream tasks. 3D scene datasets are tiny compared to the 2D field because 3D point cloud labeling is time-consuming and labor-intensive, which dramatically impedes the improvements of supervised methods. Hence many works [23, 39, 55, 57, 67, 69] explore pre-training models out of 3D labeled data to transfer knowledge for downstream tasks. The goal of self-supervised learning can be summarized as learning rich representations from unlabeled data and helping to improve performance on downstream tasks with labeled data. Most existing works follow the paradigm in the previous

*ACM MM, 2024, Melbourne, Australia*
© 2024 Copyright held by the owner/author(s). Publication rights licensed to ACM.
ACM ISBN 978-x-xxxx-xxxx-x/YY/MM
https://doi.org/10.1145/nnnnnnn.nnnnnnn

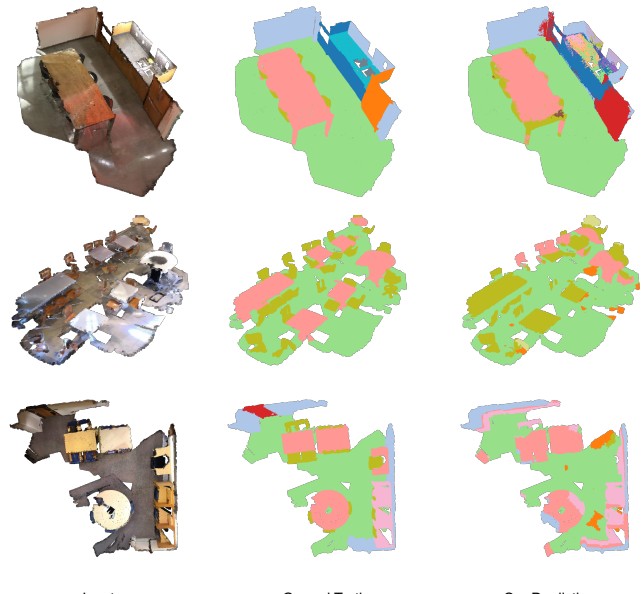

**Figure 1: The visualization of unsupervised semantic segmentation results. For better visualization, we use the Hungarian matching alignment to project the pseudo-labels to ground-truth labels.**

2D field, such as contrastive learning [26, 27, 57, 70] and masked autoencoder (MAE) [36, 39, 63, 68]. After standing on the shoulders of giants in the 2D field, we could further see the particularity of 3D representation learning as follows:

- **Unique information.** 3D scene point cloud contains various information such as geometry and color, which makes 3D point cloud data different from 2D image data. Most existing methods [26, 57, 70] treat all information of each point as an entirety in model architecture design. We argue that directly concatenating all information can not adapt the model to discriminately learn different aspects of point clouds. Although some works [52, 62] propose the two-stream architecture that encodes point cloud by 3D network and images by 2D network, it needs extra 2D data, and 3D network can not clearly learn the discrimination between different information. Considering these additional differences may be beneficial for effective representation learning.

- **Mismatch between pretraining and downstream tasks.** Previous pre-training works [26, 36, 57, 68] design their self-supervised point-level tasks, such as contrast and reconstructing between specific points. However, 3D scene downstream tasks mostly focus the object representations such as object detection and instance segmentation. The gap in supervision level

**Figure 2: Overview of our Point-GCC framework. Point-GCC utilizes the Siamese network to extract the features of geometry and color with positional embedding respectively. Then we implement the hierarchical supervision on extracted features which contains point-level *contrast and reconstruct* and object-level *contrast* based on the deep clustering module.**

between pre-training and downstream tasks may hinder the improvements of 3D self-supervised learning.

- **Architecture diversity.** The 3D point cloud field has grown rapidly in recent years [16, 33, 38, 42, 47], and the popular architecture appears changeable and specific for downstream tasks. Hence a universal pre-training framework is important that can implement various existing methods for all kinds of tasks and is easy to adapt for future architecture.

To mitigate the aforementioned problems, we explore a 3D self-supervised paradigm that can better utilize the relations of point cloud information. Most 3D scene datasets [2, 19, 21, 49, 72] provide geometry and color information, representing different aspects of the point cloud. Geometry information describes the outline of objects and can easily distinguish between them, while color information refines the internal characteristics of objects and gives a more accurate view of each object. What's more, different information has inherent relevance. For instance, we can roughly infer the geometric structure of the object from a color photo and vice versa. Motivated by the difference and relevance inherent in the information, we propose a self-supervised 3D scene pre-training framework via **G**eometry-**C**olor **C**ontrast (Point-GCC), which uses a Siamese network to extract representations and implements elaborate hierarchical supervision. To bridge the gap between pre-training and downstream tasks, the hierarchical supervision contains point-level supervision that aims to align point-wise features and object-level supervision based on a novel deep clustering module to provide better object-level representations strongly associated with downstream tasks. Additionally, the universal Siamese network is designed as an architecture-agnostic backbone so that various downstream models can easily be adapted in a plug-and-play way.

In extensive experiments, we directly perform a fully unsupervised semantic segmentation task without fine-tuning to evaluate

the quality of the pre-training model. The result outperforms the previous method with +7.8% mIoU on ScanNetV2, which proves that Point-GCC has learned rich object representations through our paradigm. Furthermore, we choose a broad downstream task to demonstrate our generality: object detection, semantic segmentation and instance segmentation on ScanNetV2 [19], SUN RGB-D [49] and S3DIS [2] datasets. Remarkably, our results indicate general improvements across all tasks and datasets. For example, we achieves new state-of-the-art results with 69.7% $AP_{25}$, 54.0% $AP_{50}$ on SUN RGB-D and 75.1% $AP_{25}$, 56.7% $AP_{50}$ on S3DIS datasets. Compared with previous pre-training methods, our method achieves higher $AP_{50}$ by +3.1% on ScanNetV2 and +1.1% on SUN RGB-D. Our contributions can be summarized as follows:

- We propose a new universal self-supervised 3D scene pre-training framework, called Point-GCC, which aligns geometry and color information via a Siamese network with hierarchical supervision. To the best of our knowledge, this is the first study to explore the alignment between geometry and color information of point cloud via the pre-training approach.
- We design a novel deep clustering module to generate object pseudo-labels based on the inherent feature consistency of the two pieces of information. The result demonstrates that Point-GCC has learned rich object representations by clustering.
- Extensive experiments show that Point-GCC is a general pre-training framework with an architecture-agnostic backbone, significantly improving performance on a wide range of downstream tasks and achieving new state-of-the-art on multiple datasets.

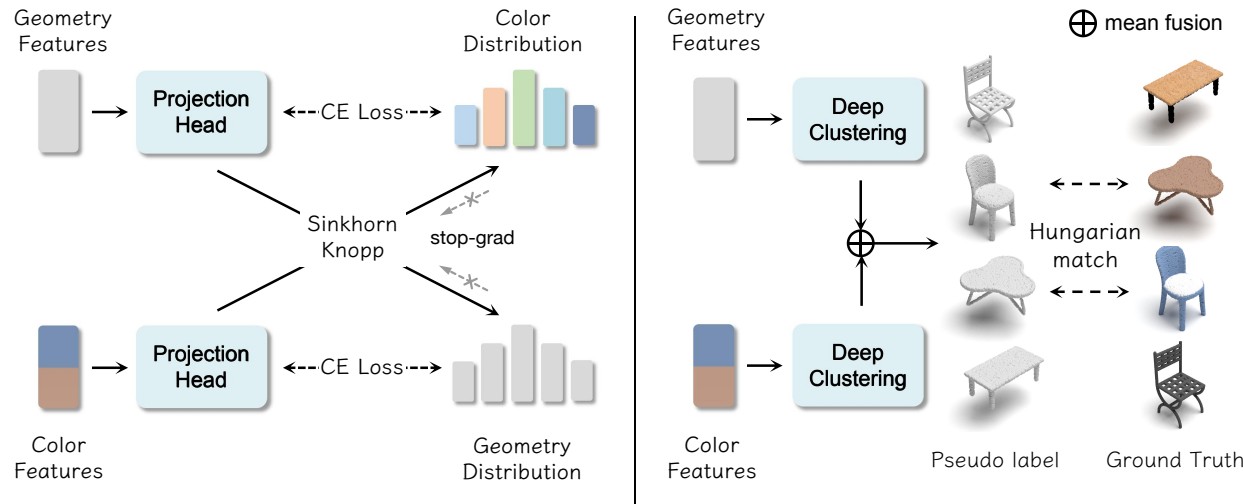

Figure 3: (a) The deep clustering module obtains pseudo prediction for different features and enforces consistent with the swapped partition distribution from the Sinkhorn-Knop algorithm. (b) Point-GCC generates the pseudo-labels by utilizing cluster prediction from both branches and projects to ground-truth labels for unsupervised semantic segmentation using Hungarian matching alignment.

## 2 RELATED WORK

### 2.1 3D Scene Understanding

Most 3D scene understanding works are still specially designed for downstream tasks, such as object detection [33, 35, 37, 44, 47, 48, 51, 52], semantic segmentation [11, 31, 38, 45, 46, 53, 66], and instance segmentation [13, 25, 29, 30, 50]. The model architecture can be summarized as a backbone module extracting the features of point clouds, and a downstream head adapting for the special task. According to the processing method, these works can be roughly divided into two categories: point-based methods and voxel-based methods. Point-based methods [11, 33, 37, 54, 71] are widely used in point clouds thanks to the effectiveness of PointNet++ [38], which alternately use farthest point algorithm and multi-layer percep-tron to sample and extract the features of point. Voxel-based meth-ods [13, 29, 30, 47, 48, 50] is recently popular because of the better performance and efficiency on many downstream tasks than point-based methods, which operate 3D sparse convolution on regular voxels transformed from irregular point clouds. We pre-train on both point-based PointNet++ and voxel-based 3D sparse convolu-tion backbone and fine-tune on multiple downstream methods to give a comprehensive view of our work.

### 2.2 3D Self-supervised Learning

Compared to 2D vision or natural language, 3D vision has a more serious problem of data scarcity [23] which limits the downstream performance of 3D tasks. To solve the raising problem, 3D self-supervised learning (SSL) [1, 28, 55, 64, 69] has gotten more atten-tion in recent years. The mainstream SSL methods can be roughly divided into two categories: contrastive learning and reconstructive learning. Contrastive learning is motivated to learn the invariant

representation from different paired carriers such as view augmen-tation [14, 15, 57] or different data formats [43, 60]. Reconstruc-tive learning is designed to reconstruct the disturbed data to learn geometry knowledge between patches [6, 22]. Motivated by the success of masked autoencoder in 2D [24, 59], the MAE-style self-supervised method became popular in point cloud [36, 41, 61, 68]. Recently, some works find that the *pattern difference* between the two methods in attention area [58] and scaling performance [39, 40]. Based on previous work, we consider the color and geometry of scene point clouds as two views for contrastive learning, and use a *swapped reconstruct* strategy for reconstructive learning. Therefore, Point-GCC achieves the integration of two methods and derives benefits from both of them.

### 2.3 Deep Clustering for Self-supervised Learning

Deep Clustering [5, 7, 9, 12, 34, 56, 65] aims to learn better fea-tures and discover data labels using deep neural networks, which has been broadly applied in self-supervised and semi-supervised learning. DeepCluster [8] uses the off-the-shelf K-means algorithm pseudo-labels as supervision which learns comparative representa-tions for self-supervised learning. SeLa [4] proposes a simultaneous clustering and representation learning method using the Sinkhorn-Knopp algorithm to generate pseudo-labels with equal partitions quickly. SwAV [10] combines contrastive learning and deep clus-tering, which enforces consistency between cluster assignments from different views of the same image. In this work, we attempt to apply deep clustering in 3D self-supervised learning field, which generates pseudo-labels based on the inherent feature consistency of the geometry and color information of the point cloud.

# 3 POINT-GCC: PRE-TRAINING VIA GEOMETRY-COLOR CONTRAST

Existing methods mainly focus on geometric information, but our goal is to enhance the 3D representation capability by better utilizing all the information discriminately in scene point clouds. Therefore, a novel *Geometry-Color Contrast* method is proposed to address this motivation. Figure 2 illustrates the overall framework of Point-GCC. We first perform a Siamese backbone to extract the features of the geometry and color information respectively in Section 3.1. To carefully align the features belonging to different information, we propose the point-level supervision via combining the contrastive and reconstructive learning in Section 3.2, then we design an unsupervised deep clustering module to generate object pseudo-labels and perform object-level contrastive learning between high-confidence object samples in Section 3.3. The final hierarchical supervision is described in Section 3.4. In Section 3.5, we propose a new method directly evaluating the pre-training model on unsupervised semantic segmentation to demonstrate the effectiveness of our method.

## 3.1 Siamese Architecture

**Information split and embedding.** In 3D scene datasets, a point $p$ is usually associated geometry information represented by the coordinates $p_{geo}$ and color information represented by RGB value $p_{color}$. Different from previous pre-training methods regarding a single point as an atom unit, we split the point cloud into two parts, the geometry and color respectively. Then we project them to universal embedding space $e$ by Equation 1. Additionally, to distinguish similar colors in different coord, we add an extra weakly positional embedding $e_{pos}$ to the color embedding with the Euclidean norm of coord. Positional embedding helps the network distinguish different objects that share the same color and improves generalization in various scenarios. Note that we remove all embedding modules in fine-tuning stage to keep our framework plug-and-play in order that more existing methods can benefit from ours.

**Siamese architecture-agnostic backbone.** We use a symmetric Siamese network $\mathcal{F}(\cdot)$ to separately encode geometry features $f_{geo}$ and color features $f_{color}$. Since we attempt to help more existing architectures learn better representations from the combination of geometry and color information, we do not modify any backbone architecture. So that we can directly reuse the core module for standard segmentation with any backbone architecture. In other words, the backbone encodes input $x \in R^{N \times C_1}$ and extracts feature $y \in R^{N \times C_2}$. To align the two information, Siamese backbone $\mathcal{F}(\cdot)$ encodes the geometry embedding $e_{geo}$ and color embedding $e_{color}$ with weakly positional embedding $e_{pos}$ to geometry features $f_{geo}$ and color features $f_{color}$ respectively:

$$e_{geo} = \mathcal{E}_{geo}(p_{geo}), \quad e_{color} = \mathcal{E}_{color}(p_{color}),$$
$$e_{pos} = \mathcal{E}_{pos}(\|p_{geo}\|_2^2), \tag{1}$$
$$f_{geo} = \mathcal{F}(e_{geo}), \quad f_{color} = \mathcal{F}(e_{color} + e_{pos}), \tag{2}$$

where $\mathcal{E}$ is corresponding linear layer of each embedding, $\mathcal{F}(\cdot)$ is the Siamese network.

In our opinion, the Siamese backbone network is forced to apply the discriminative distributions and learn the relation and distinction of different attributions from the same points. The learned knowledge is critical for transferring to all discriminative inputs concatenated situations in most downstream tasks.

## 3.2 Point-level Supervision

Inspired by the success of associating contrastive learning and reconstructive learning in recent work [39], We propose our point-level supervision elaborately designed for our Siamese architecture, which first contrasts and then *swapped reconstruct* the features to benefit from different paradigms.

**Contrastive learning.** The geometry features $f_{geo}$ and color feature $f_{color}$ are point-wise aligned because they are split from the same point cloud $p$ and extracted by the Siamese segmentation-style backbone network. We apply the InfoNCE loss aiming to pull positive pairs close, and push negative pairs away across the geometry features and color features:

$$\mathcal{L}_{pc} = -\sum_i^N \log \frac{\exp\left(z_{geo}^{iT} \cdot z_{color}^i / \tau\right)}{\sum_j^N \exp\left(z_{geo}^{iT} \cdot z_{color}^j / \tau\right)}, \tag{3}$$

where $\tau$ is the temperature hyper-parameter, we follow the previous works [57] to set it as 0.4. $z_{geo}^i$ and $z_{color}^i$ correspond to matched $\ell_2$-normalized feature $f_{geo}^i$ and $f_{color}^i$ from same point $p^i$, which represent a pair of positive sample. And $z_{geo}^i$ with other $z_{color}^j$ except $z_{color}^i$ represent negative pairs.

**Reconstructive learning.** Based on our Siamese architecture, we apply the reconstructive learning by *swapped reconstruct* strategy instead of mask strategy, which solves the raising problem about the distribution mismatch between training and testing data in masked autoencoding for point cloud [32]. Specifically, we simply project the geometry features $f_{geo}$ and color features $f_{color}$ to reconstruct color $\hat{p}_{geo}$ and geometry $\hat{p}_{color}$. The reconstructive loss is the mean squared error (MSE) between the reconstructed and original information of each point:

$$\mathcal{L}_{pr} = \frac{1}{N} \sum \|p_{geo}^{i\prime} - \hat{p}_{geo}^i\|_2^2 + \frac{1}{N} \sum \|p_{color}^{i\prime} - \hat{p}_{color}^i\|_2^2, \tag{4}$$

where $N$ is the number of points, $\hat{p}_{geo}^i$ and $\hat{p}_{color}^i$ represent the reconstruct prediction, $p_{geo}^{i\prime}$ and $p_{color}^{i\prime}$ represent the reconstruct targets which both scale to between 0 and 1 for stability training loss.

## 3.3 Object-level Supervision

Point-level supervision is widely applied in 3D self-supervised learning, which provides rich representations for downstream tasks. However, the object representation strongly associated with downstream tasks hasn't been noticed before. We propose our object-level supervision driven by the novel unsupervised deep clustering module. The clustering module generates pseudo-label predictions $\mathcal{P}_{geo}$ and $\mathcal{P}_{color}$ for the geometry features $f_{geo}$ and color features $f_{color}$ respectively, and enforces consistent prediction between geometry prediction $\mathcal{P}_{geo}$ and color prediction $\mathcal{P}_{color}$ of same point $p$. We argue that the pseudo-labels represent more various object features, which are not restricted by human annotations with fixed

**Table 1: 3D Object detection results on ScanNetV2 [20], SUN RGB-D [49] validation set. The overall best results are bold, and the best results with the same baseline model are underlined. * means that we evaluate on VoteNet [37] with the stronger MMDetection3D [17] implementation.**

| Method | ScanNetV2 | | SUN RGB-D | |
|---|---|---|---|---|
| | $AP_{25}$ | $AP_{50}$ | $AP_{25}$ | $AP_{50}$ |
| **Supervised Only** | | | | |
| VoteNet [37] | 58.6 | 33.5 | 57.7 | - |
| GroupFree-3D [33] | 66.3 | 47.8 | - | - |
| FCAF3D [47] | 71.5 | 57.3 | 64.2 | 48.9 |
| TR3D [48] | 72.9 | 59.3 | 67.1 | 50.4 |
| **Self-supervised Pre-training** | | | | |
| VoteNet [37] | 58.6 | 33.5 | 57.7 | - |
| + PointContrast [57] | 59.2 | 38.0 | 57.5 | 34.8 |
| + DepthContrast [70] | 62.1 | 39.1 | 60.4 | 35.4 |
| + CSC [26] | - | 39.3 | - | 36.4 |
| + Ponder [27] | 63.6 | 41.0 | 61.0 | 36.6 |
| + Point-GCC* | 65.3 (+3.0) | 44.1 (+3.3) | 61.3 (+1.5) | 37.7 (+2.0) |
| VoteNet+FF [48] | - | - | 64.5 | 39.2 |
| + Point-GCC | - | - | 64.9 (+0.4) | 41.3 (+2.1) |
| GroupFree-3D [33] | 66.3 | 47.8 | - | - |
| + Point-GCC | 68.1 (+1.8) | 49.2 (+1.4) | - | - |
| TR3D [48] | 72.9 | 59.3 | 67.1 | 50.4 |
| + Point-GCC | 73.1 (+0.2) | 59.6 (+0.3) | 67.7 (+0.6) | 51.0 (+0.6) |
| TR3D+FF [48] | - | - | 69.4 | 53.4 |
| + Point-GCC | - | - | 69.7 (+0.3) | 54.0 (+0.6) |

object classes. To achieve robust supervision among these object-level pseudo labels, we sample the high-confidence object features based on the prediction confidence score and apply object-level contrastive learning according to pseudo labels.

**Deep clustering via swapped prediction.** We apply the swapped prediction [10] in 2D contrastive learning to our model, which predicts the pseudo label of an image from the clustering result of another view. In our framework, we swap the cluster target of different information features, and predict the pseudo label from the other information feature based on the inherent consistency of the two types of information as shown in Figure 3(a). For pseudo label classes $K$, we use a learnable matrix $C = [c_1, \cdots, c_K]$ to represent the cluster centroids, and calculate the similarity $S$ between the $\ell_2$-normalized features $f$ and cluster centroids $c$. To avoid the degeneration problem that all features collapse into the same prediction, the Sinkhorn-Knopp algorithm [18] is used to generate the equal partition cluster distribution $Q$ from the similarity $S$ by converting pseudo-label generation to an optimal transport problem. And the learnable prediction $\mathcal{P}$ is computed by $softmax(S/\tau)$, where $\tau$ is the temperature hyper-parameter. We set all hyper-parameter in swapped prediction same to the previous works [10] in 2D. Finally, The swapped prediction loss is the cross entropy losses between the learnable prediction $\mathcal{P}$ and swapped equal partition distribution $Q$:

$$\mathcal{L}_{clu} = \ell(Q_{geo}, \mathcal{P}_{color}) + \ell(Q_{color}, \mathcal{P}_{geo}), \quad (5)$$

**Table 2: 3D Object detection results on S3DIS [3] validation set. $\dagger$ means with extra training dataset ScanNetV2 [20].**

| Method | S3DIS | |
|---|---|---|
| | $AP_{25}$ | $AP_{50}$ |
| **Supervised Only** | | |
| FCAF3D [47] | 66.7 | 45.9 |
| TR3D [48] | 74.5 | 51.7 |
| **Self-supervised Pre-training** | | |
| TR3D [48] | 74.5 | 51.7 |
| + Point-GCC | 74.9 (+0.4) | 53.2 (+1.5) |
| + Point-GCC$^\dagger$ | 75.1 (+0.6) | 56.7 (+5.0) |

where $\ell$ is the cross-entropy loss between the prediction and target. **Object-level contrastive learning.** For the features $f$ with corresponding pseudo prediction $\mathcal{P}$ and confidence score from deep clustering, we pick features with confidence scores higher than the picking threshold to alleviate the noise from unsupervised clustering. Then we compute the mean features of high-confidence samples from geometry and color branches, respectively. We take the two types of mean features with the same pseudo-label as positive pairs, oppositely with different pseudo-label as negative pairs, and apply the InfoNCE loss at object-level:

$$\mathcal{L}_{oc} = - \sum_i^N \log \frac{\exp\left(z_{geo}^{iT} \cdot z_{color}^i / \tau\right)}{\sum_j^N \exp\left(z_{geo}^{iT} \cdot z_{color}^j / \tau\right)}, \quad (6)$$

where $\tau$ is the temperature hyper-parameter, we set it to 0.4 following the above-mentioned setting. $z^i$ is the $\ell_2$-normalized mean feature with pseudo-label $i$. $z_{geo}^i$ and $z_{color}^i$ represent a pair of positive sample with same pseudo-label $i$. And $z_{geo}^i$ with $z_{color}^j$ corresponding different pseudo-label $j$ represent negative samples. We believe the pseudo-labels represent various object features that are not restricted by human annotations with fixed object classes and are significant for learning rich representations.

## 3.4 Overall Hierarchical Loss

Our framework contains hierarchical supervision at point-level and object-level, and the final loss is a combination of the four losses above-mentioned:

$$\mathcal{L}_{over} = \mathcal{L}_{pc} + \alpha\mathcal{L}_{pr} + \beta\mathcal{L}_{clu} + \gamma\mathcal{L}_{oc}, \quad (7)$$

where $\alpha$, $\beta$ and $\gamma$ are the loss weight hyper-parameters, we set them to 100, 100 and 1 respectively to balance the magnitude of losses.

## 3.5 Adapt to unsupervised semantic segmentation

Due to the pseudo-label from object-level supervision, Point-GCC can adapt to unsupervised downstream tasks without fine-tuning. Meanwhile, previous pre-training methods evaluate the performance by transfer learning on downstream tasks. The results can be greatly affected by the fine-tuning setting and are not intuitive between different baselines. As shown in Figure 3(b), we generate

Table 3: 3D semantic segmentation results on ScanNetV2 [20] dataset by different level of supervision. The overall best results are bold. + means fine-tuning with pre-training on the corresponding dataset.

| Method | Supervision | Backbone | Pseudo Classes | mIoU |
|---|---|---|---|---|
| **Unsupervised Method** | | | | |
| SL3D [11] | unsupervised | PointNet++ | 400 | 8.5 |
| SL3D [11] | unsupervised | Point Transformer | 800 | 10.5 |
| Point-GCC | unsupervised | PointNet++ | 20 | 18.3 |
| **Weakly-supervised Method** | | | | |
| WyPR [45] | scene-level | PointNet++ | 20 | 29.6 |
| MPRM [53] | subcloud-level | KPConv | 20 | 41.0 |
| **Supervised Method** | | | | |
| PointNet++ [38] | supervised | PointNet++ | 20 | 54.4 |
| + Point-GCC | supervised | PointNet++ | 20 | **59.8** |

the final pseudo-labels by utilizing cluster prediction from geometry and color branch. During the evaluation stage, we use the Hungarian matching alignment [65] to project the pseudo-labels to ground-truth labels because we are agnostic to the ground truth in pre-training. Although our method is not specifically designed for unsupervised downstream tasks, we find that the process is more intuitive and fair for evaluating the performance of pre-training methods.

## 4  EXPERIMENTS

To analyze the 3D representation learned by Point-GCC, we conduct extensive experiments on multiple datasets and tasks described in Section 4.1. First, we evaluate fully unsupervised semantic segmentation tasks to validate the effectiveness of object representation in Section 4.2. Then we expand experiments by transfer learning on multiple downstream tasks and datasets in Section 4.3.

### 4.1  Experiment setting

**Dataset.** We use three popular indoor scene datasets: ScanNetV2 [19], SUN RGB-D [49], S3DIS [2] in our experiments. **ScanNetV2** is a 3D reconstruction dataset, which provides 1513 indoor scans with a total of 20 classes. **SUN RGB-D** is a monocular RGB-D image dataset, which provides 10335 RGB-D images from four different sensors with a total of 37 classes. **S3DIS** is another 3D indoor scene dataset, which provides 271 point cloud scenes across 6 areas with 13 classes.

**Implementation details.** We implement Point-GCC built upon the MMDetection3D [17] framework. We use the AdamW optimizer with an initial learning rate of 0.001 and weight decay of 0.0001. Other implementation details follow the default scheme. To ensure fair comparability of results, we refer to selecting downstream models implemented by MMDetection3D. In downstream task experiments, we decay the learning rate by 0.5, and other settings follow the original implementation. The full detail settings are provided in the Appendix.

Table 4: 3D instance segmentation results on ScanNetV2 [20] and S3DIS [3] dataset. The overall best results are bold, and the best results with the same baseline model are underlined. + means fine-tuning with pre-training on the corresponding dataset. $^{\dagger}$ means with extra training dataset ScanNetV2 [20].

| Method | ScanNetV2 | | | S3DIS | | | |
|---|---|---|---|---|---|---|---|
| | AP | $AP_{50}$ | $AP_{25}$ | AP | $AP_{50}$ | $Prec_{50}$ | $Rec_{50}$ |
| **Supervised Only** | | | | | | | |
| PointGroup [29] | 34.8 | 56.7 | 71.3 | - | 57.8 | 61.9 | 62.1 |
| HAIS$^{\dagger}$ [13] | 43.5 | 64.4 | 75.6 | - | - | 71.1 | 65.0 |
| SoftGroup$^{\dagger}$ [50] | 45.8 | 67.6 | 78.9 | 51.6 | 66.1 | 73.6 | 66.6 |
| **Self-supervised Pre-training** | | | | | | | |
| TD3D [30] | 46.2 | 71.1 | 81.3 | 48.6 | 65.1 | 74.4 | 64.8 |
| + Point-GCC | 47.3 | 71.3 | 81.6 | 50.5 | 65.4 | 75.5 | 65.9 |
| TD3D$^{\dagger}$ [30] | - | - | - | 52.1 | 67.2 | 75.2 | 68.7 |
| + Point-GCC$^{\dagger}$ | - | - | - | **53.6** | **68.4** | **76.6** | **69.5** |

### 4.2  Fully unsupervised semantic segmentation

We evaluate our pre-training model on fully unsupervised semantic segmentation tasks using the method in Section 3.5 to validate the effectiveness of object representation. As shown in Table 3, our method surpasses previous unsupervised methods by a huge margin and is closer to the weakly-supervised method, despite Point-GCC being not specifically designed for unsupervised downstream tasks. With the same backbone PointNet++, Point-GCC surpasses previous work SL3D [11] by +9.8% mIoU, and +7.8% mIoU compared with more powerful Point Transformer on ScanNetV2 dataset. The result proves that Point-GCC has learned rich object representation in unsupervised pre-training.

**Fine-tuning semantic segmentation.** Additionally, we fine-tune the pre-training model for semantic segmentation to verify the consistent improvement of our method. With supervised fine-tuning, the model gains significant improvements by +5.4% mIoU on the ScanNetV2 dataset, which proves that our method has learned intrinsic representations of the point cloud.

### 4.3  Transfer learning on downstream tasks

**3D Object detection.** For 3D object detection task, we pre-train the PointNet++ [38] backbone for VoteNet [37], VoteNet+FF [48] and GroupFree-3D [33] and the MinkResNet [16] backbone for TR3D [48], TR3D+FF [48] respectively. Table 1 and 2 shows the results on ScanNetV2, SUN-RGBD, and S3DIS datasets. Our method gains stable and significant improvements for various settings. Compared with previous 3D self-supervised methods with the common baseline model VoteNet, our method achieves higher $AP_{50}$ than the previous highest model Ponder [27] by +3.1% on ScanNetV2 and +1.1% on SUN RGB-D. For more recent models, our model also significantly boosts VoteNet+FF, GroupFree-3D, TR3D, TR3D+FF on multiple datasets and achieves new state-of-the-art results with 69.7% $AP_{25}$, 54.0% $AP_{50}$ on SUN RGB-D and 75.1% $AP_{25}$, 56.7% $AP_{50}$ on S3DIS datasets.

**3D Instance segmentation.** For 3D instance segmentation task, we pre-train the MinkResNet backbone for TD3D [30] on ScanNet

**Table 5: Additional detection comparison with different information on ScanNetV2 [20] validation set. * means the VoteNet with a stronger MMDetection3D [17] implementation.**

| Method | Input | Object Detection | |
|---|---|---|---|
| | | $AP_{25}$ | $AP_{50}$ |
| VoteNet [37] | xyz+height | 58.6 | 33.5 |
| VoteNet* | xyz+height | 62.3 | 40.8 |
| VoteNet* | xyz+color | 61.8 | 39.9 |
| + Point-GCC* | xyz+color | 65.3 (+3.0) | 44.1 (+3.3) |
| GroupFree-3D [33] | xyz | 66.3 | 47.8 |
| GroupFree-3D [52] | xyz+color | 66.3 | 47.0 |
| + Point-GCC | xyz+color | 68.1 (+1.8) | 49.2 (+1.4) |

**Table 6: Ablation study of the hierarchical supervision. - means the model can't perform the unsupervised segmentation task due to the lack of the pseudo-label.**

| Point-level | | Object-level | | Unsupervised Segmentation | Object Detection | |
|---|---|---|---|---|---|---|
| Contra. | Recon. | Cluster. | Contra. | mIoU | $AP_{25}$ | $AP_{50}$ |
| ✗ | ✗ | ✗ | ✗ | - | 62.3 | 40.8 |
| ✗ | ✓ | ✗ | ✗ | - | 63.3 | 42.7 |
| ✓ | ✗ | ✗ | ✗ | - | 64.4 | 42.8 |
| ✓ | ✓ | ✗ | ✗ | - | 64.8 | 43.0 |
| ✓ | ✓ | ✓ | ✗ | 16.07 | 65.0 | 43.6 |
| ✓ | ✓ | ✓ | ✓ | **18.27** | **65.3** | **44.1** |

and S3DIS datasets. Table 4 shows the results on ScanNetV2 and S3DIS validation sets. Downstream models gain remarkable performance by +1.1% AP on ScanNetV2, +1.9% on S3DIS and +1.5% on S3DIS with extra train data, demonstrating our method's general improvement across multiple settings.

Interestingly, the improvements for the PointNet++ [38] backbone widely surpass the MinkResNet [16] backbone. We guess that sparse convolution architecture implicitly aligns the color information from features and the geometry information from fine-grained sparse voxel operation. It may be a kind of explanation for why 3D sparse convolution has better performance and efficiency on various tasks.

**Additional comparison** Some baseline models [33, 37] do not use color information. Table 5 shows the additional baseline model with color information for a fair comparison from our reproduction and other works [27, 52], which get slight improvement and even decrease. The results prove what we mentioned in the main paper: directly concatenating all information can not adapt the model to learn different aspects of point clouds discriminately, demonstrating our work's necessity.

## 4.4 Ablation study And Discussion

To analyze the effectiveness of our approach, we further explore additional experiments to measure the contribution of each component to the final representation quality. For efficiency, all ablation experiments are implemented with VoteNet setting on pre-training and object detection.

**Hierarchical supervision.** To further explore the improvement of our hierarchical supervision, we conduct ablation studies with different components. Table 6 shows the unsupervised semantic segmentation results with pre-training and object detection results with fine-tuning. The results show that both contrastive learning and reconstructive learning at point-level contribute to the final results. Even though just with point-level supervision, our method has achieved higher $AP_{25}$ and $AP_{50}$ than the previous best model Ponder by +1.2% and +2.0%. Furthermore, the swapped prediction and object-level contrastive learning also provide remarkable improvements for $AP_{50}$ and $AP_{25}$, especially $AP_{50}$. Intuitively, the

improvement of $AP_{50}$ is more significant than $AP_{25}$, demonstrating that object-level supervision improves the model with a more precise view of objects.

**Weakly positional embedding** The color information is vague for precise geometry reconstruction, and some backbone networks are translational equivariant and hard to predict a global coordinate only with color information embedding, i.e., some different objects partly share the same color can not be distinguished. The weakly positional embedding is designed to add to color information embedding to provide blurry global position information. Positional embedding provides implicit global information to help reconstruct precise geometry from vague color information, which balances the task difficulty between two branches. It also helps the network distinguish different objects that share the same color and improves generalization among various scenarios.

We conduct an additional ablation study with different settings to explore the improvement of positional embedding further. Table 7 shows the object detection results with different positional embedding. **no pos** means without positional embedding. **xy pos** means with the positional embedding of x and y, and the corresponding task aims only to reconstruct z, *i.e.*, height estimation task. The results show that our model has stable effects under different position encoding conditions. This may be because we remove the positional embedding and adjust the input channel in the downstream fine-tuning stage so that the impact on the downstream task is reduced.

**Geometry-Color Contrast.** To verify the importance of Geometry-Color Contrast approach, we compare the results with a single reconstruction branch setting. Table 8 shows object detection results with different pre-training branches. The results show that the performance with a single branch of whether geometry or color reconstruction obviously declines, which proves our Geometry-Color Contrast plays an essential role in the significant performance.

**Object sampling strategy.** The result in table 6 shows that object-level supervision provides the most obvious boost for $AP_{50}$. We compare the results with different object sampling strategies to analyze the object samples used in object-level contrastive learning. The results in table 9 show that the more confident object samples are, the greater performance we achieve. However, the performance decays only using the maximum score sample because of overfitting.

**Table 7: Ablation study of the positional embedding. xy pos means only x and y are used for positional embedding.**

| Positional Embedding | Object Detection | |
| --- | --- | --- |
| | $AP_{25}$ | $AP_{50}$ |
| no pos | 64.5 | 43.7 |
| xy pos | 64.7 | 43.8 |
| norm pos | **65.3** | **44.1** |

**Table 8: Ablation study of the Geometry-Color Contrast approach.**

| Color Branch | Geometry Branch | Object Detection | |
| --- | --- | --- | --- |
| | | $AP_{25}$ | $AP_{50}$ |
| ✗ | ✗ | 62.3 | 40.8 |
| ✓ | ✗ | 62.5 | 39.4 |
| ✗ | ✓ | 62.4 | 40.9 |
| ✓ | ✓ | **64.8** | **43.0** |

**Table 9: Ablation study of the Object sampling strategy.**

| Object Picking Threshold | Object Detection | |
| --- | --- | --- |
| | $AP_{25}$ | $AP_{50}$ |
| 1.5 / class num. | 64.2 | 43.2 |
| 1.8 / class num. | 64.4 | 43.7 |
| 2.0 / class num. | **65.3** | **44.1** |
| only max score | 64.5 | 43.0 |

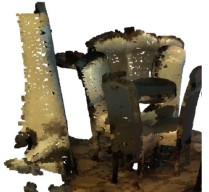 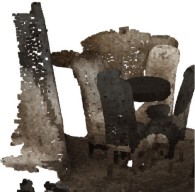 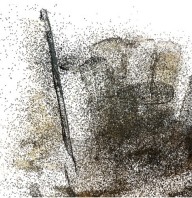

Ground Truth     Color Reconstruction     Geometry Reconstruction

**Figure 4: The visualization of reconstruction results from Point-GCC. Note that we decrease the point size in geometry reconstruction to avoid the block from noisy points.**

1.8 /class      2.0 /class      2.2 /class

**Figure 5: The visualizations of used points after the different thresholds.**

## 4.5 Visualization

**Unsupervised semantic segmentation visualization** We provide visualization results of unsupervised semantic segmentation. Figure 1 shows that our method can clearly distinguish the main parts of different objects without supervision. However, for small or complex objects, the segment may be merged into others or ignored because of the equal partition cluster distribution from the Sinkhorn-Knopp algorithm [18]. The result demonstrates that our pre-training approach helps the model learn object representations to enhance performance on downstream tasks.

**Object sampling strategy visualization** From the following experiments, we can find that the quality of points used for object-level contrastive learning is important for performance. We believe that the quality of object-level features depends on the trade-off between quantity and quality of point-level features. For a more intuitive perspective, we visualize points used after different thresholds in Figure 5. Shrinking the threshold helps retrieve more accurate points that belong to the same semantic class, but the lack of quantity makes it difficult to reveal the essential object-level feature. However, simply relaxing the threshold will introduce more noise that belongs to different semantic classes, inevitably degrading object-level feature quality.

**Geometry and color reconstruction visualization** Figure 4 shows the visualization of geometry and color reconstruction results. The results show that our method can consistently generate high-quality complements from one type of information in the point cloud. The method may contain potential applications such as depth estimation and texture generation.

## 5 FUTURE WORK

As mentioned above, most of the limitations come from our compromise to general adaptation. Global coordinates make it easy for our work to adapt to different model architectures and datasets but may also cause the distribution mismatch problem. The preferred aim of this work is to help improve various existing downstream methods. Our future exploration will consider more elaborate architectural designs for better point cloud understanding and performance, such as geometry and color information embedding and pre-training tasks.

## 6 CONCLUSIONS

In this paper, we propose a new universal self-supervised 3D scene pre-training framework via **G**eometry-**C**olor **C**ontrast (Point-GCC), which utilizes an architecture-agnostic Siamese network with hierarchical supervision. Extensive experiments show that Point-GCC significantly improves performance on unsupervised tasks without fine-tuning and a wide range of downstream tasks, especially achieving new state-of-the-art results on multiple datasets.

To the best of our knowledge, Point-GCC is the first study to explore the self-supervised paradigm that can better utilize the relations of different point cloud information, hence we elaborately design our plug-and-play pre-training framework to help improve various existing downstream methods, instead of directly designing a new architecture. We hope our work could attract more attention to the discriminative information of point clouds, which may inspire future point cloud representation learning works.

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
