# OpenReview forum: "Point-GCC: Universal Self-supervised 3D Scene Pre-training via Geometry-Color Contrast"
_acmmm.org/ACMMM/2024/Conference — MM2024 Poster_

### Official Review · Reviewer_ySvA · 2024-05-05

**Rating:** 4
**Confidence:** 3

**Summary:**

This paper introduces a novel self-supervised 3D scene pre-training framework. It aligns geometry and color information using a Siamese network with point-level and object-level supervisions. Additionally, it proposes a deep clustering module to generate object pseudo-labels. The effectiveness of the framework is demonstrated through experiments on three datasets and three different downstream tasks.

**Strengths:**

1. The paper is written clearly, making it easy for the reviewer to understand.
2. The experiments in Table 1 are impressive, as they compare the latest methods and test various backbones.
3. The ablation study is clear and effectively presents the contribution of each component.

**Limitations:**

1. In the experiments outlined in Section 4.2, the comparisons appear to be insufficient. Specifically, for the unsupervised semantic segmentation task on ScanNet V2, the authors only compared with SL3D, which is not considered state-of-the-art. For instance, GrowSP [1] achieves an mIoU of 25.4 in this task, significantly higher than the reported result. Similarly, for the fully supervised fine-tuning task, no comparison with state-of-the-art methods was provided. It is recommended to include at least one state-of-the-art method for a fair comparison to demonstrate the effectiveness of this method.
2. Regarding the instance segmentation task discussed in Section 4.3, it is encouraged to include a comparison with at least one state-of-the-art method.
3. There are concerns about the fairness of the comparison to Ponder on the 3D detection task in Table 1. The VoteNet used in this paper achieves a much higher AP than the officially reported results. Is this higher performance due to a significantly higher baseline?

Ref:
[1] Z. Zhang, B. Yang, B. Wang, 和 B. Li，“Growsp: Unsupervised semantic segmentation of 3d point clouds,” in Proceedings of the IEEE/CVF Conference on Computer Vision and Pattern Recognition, 2023, pp. 17619–17629.

**Suitability:**

3

---

### Official Review · Reviewer_1H27 · 2024-05-11

**Rating:** 4
**Confidence:** 4

**Summary:**

The paper introduces a novel 3D self-supervised learning framework for pre-training on 3D scene understanding tasks. The authors propose Point-GCC, which utilizes a Siamese network to align geometry and color information from point clouds. The framework includes hierarchical supervision with both point-level and object-level contrast, and it is designed to be architecture-agnostic to accommodate various downstream models. Experiments demonstrate that Point-GCC achieves significant improvements on a range of tasks across different datasets, including new state-of-the-art results for object detection on SUN RGB-D and S3DIS datasets.

**Strengths:**

1. Innovative Approach: The paper presents a unique paradigm for 3D self-supervised learning by contrasting geometry and color information, which is a novel contribution to the field.
2. Architecture-Agnostic Design: The proposed framework is designed to be adaptable to various downstream models, making it a versatile tool for different applications.
3. Hierarchical Supervision: The use of both point-level and object-level supervision provides a more nuanced approach to learning from unlabeled data.
4. State-of-the-Art Results: The framework demonstrates excellent performance, achieving new state-of-the-art results on standard benchmarks.
5. Extensive Evaluation: The authors have conducted a thorough set of experiments, including unsupervised tasks and transfer learning to downstream tasks, which substantiates the effectiveness of the framework.

**Limitations:**

1. Complexity: The framework's design might introduce additional complexity compared to simpler models, which could potentially impact its scalability.
2. Generalization to Other Domains: While the paper shows strong results for indoor scenes, it's unclear how well the framework generalizes to other domains like outdoor scenes or different types of 3D data.
3. Computational Resources: The paper does not discuss the computational requirements or runtime, which could be a concern for researchers with limited resources.
4. Explainability and Transparency: The paper could benefit from a deeper analysis of what the model learns during the pre-training phase and how this knowledge transfers to downstream tasks.

**Suitability:**

3

---

### Official Review · Reviewer_idXX · 2024-05-24

**Rating:** 4
**Confidence:** 1

**Summary:**

The paper proposes a novel self-supervised learning framework designed to improve 3D scene understanding by leveraging both geometric and color information in point clouds. The authors introduce a Siamese network that aligns these two types of data through hierarchical supervision, including point-level contrastive and reconstructive learning as well as object-level contrastive learning facilitated by a deep clustering module. The approach is architecture-agnostic, enabling it to adapt to various downstream models. The experimental results demonstrate significant improvements across multiple datasets and tasks, such as object detection and semantic segmentation, achieving new state-of-the-art performances.

**Strengths:**

1.The introduction of a Siamese network to align geometric and color information is a novel approach that effectively utilizes the unique characteristics of 3D point cloud data.

2.The combination of point-level and object-level supervision ensures comprehensive learning of features, addressing the gap between pre-training and downstream tasks.

**Limitations:**

1.The research content of this article is limited to indoor scenes and does not mention outdoor autonomous driving scenes, so naming it "3D Scene Pre training" is somewhat inappropriate.

2.In Point level Supervision, using the color and geometric features of each point as the reconstruction target, without fusing the features of other points, can this method be successfully trained?

3.The approach assumes the availability of both geometric and color information in point cloud datasets, which may not always be the case in real-world scenarios.

4.While the paper reports improvements in performance, it lacks a detailed analysis of the computational cost and efficiency of the proposed methods compared to baseline models.

**Suitability:**

2

---

### Official Review · Reviewer_FE1o · 2024-05-24

**Rating:** 3
**Confidence:** 3

**Summary:**

This paper proposes a self-supervised 3D scene pre-training framework via Geometry-Color Contrast (Point-GCC), which utilizes an architecture-agnostic Siamese network with hierarchical supervision.

**Strengths:**

The authors design a deep clustering module to generate object pseudo-labels based on the inherent feature consistency of the two pieces of information.

**Limitations:**

1) The motivation for this work is not well discussed in both the abstract and introduction sections. What's more, The logic presented in the introduction is difficult to follow.
2) Several baseline methods selected for comparison are outdated and may not reflect recent advancements in the field.
3) The ablation study of the object sampling strategy is confusing.
4) The layout of the paper appears disorderly and could benefit from better organization and visual presentation.

**Suitability:**

2

---

### Meta-Review · Area_Chair_mdEQ · 2024-07-03

**Recommendation:** Accept (Poster)
**Confidence:** 5

**Metareview:**

This paper was reviewed by four experts in the field. The recommendations are (Borderline Accept x 4). Based on the reviewers' feedback, the decision is to recommend the acceptance of the paper. The reviewers did raise some valuable concerns (especially more detailed ablations experiments, numerical results for model efficiency and computational cost, more comparisons with related works, remove of overclaim statements, etc.) that should be addressed in the final camera-ready version of the paper. The authors are encouraged to make the necessary changes to the best of their ability. We congratulate the authors on the acceptance of their paper.